# Investigation of Microstructure of Al-5Ti-0.62C System and Synthesis Mechanism of TiC

**DOI:** 10.3390/ma13020310

**Published:** 2020-01-09

**Authors:** Wanwu Ding, Taili Chen, Xiaoyan Zhao, Yan Cheng, Xiaoxiong Liu, Lumin Gou

**Affiliations:** 1State Key Laboratory of Advanced Processing and Recycling of Nonferrous Metals, Lanzhou University of Technology, Lanzhou 730050, China; chen_taili@163.com (T.C.); zxyxmy@163.com (X.Z.); chengyanLUT@163.com (Y.C.); lxxlbj@163.com (X.L.); goulumin@163.com (L.G.); 2School of Materials Science and Engineering, Lanzhou University of Technology, Lanzhou 730050, China

**Keywords:** Al-5Ti-0.62C master alloy, quenching experiment, grain refining, differential scanning calorimeter

## Abstract

Al-Ti-C master alloys have been widely investigated by various researchers. However, their refining effectiveness is still severely compromised by the preparation process. In this work, the aluminum melt in-situ reaction was carried out to synthesize the Al-5Ti-0.62C, and its refining performance was estimated. The thermodynamics calculation and differential scanning calorimeter experiment were used to investigate the synthesis mechanism of TiC. Quenching experiment was conducted to explore phase and microstructure transformation of the Al-5Ti-0.62C system. The results show that the main phases of Al-5Ti-0.62C master alloys are α-Al, Al_3_Ti, and TiC and it has a positive effect on commercial pure aluminum refining. Commercial pure aluminum is completely refined into the fine equiaxed structure by adding 0.3% Al-5Ti-0.62C master alloy. TiC particles mainly distribute in the grain interior and grain boundaries. The excess Ti came from the dissolution of Al_3_Ti spreading around TiC and finally forming the Ti-rich zone to promote the nucleation of α-Al. The experiments certified that TiC was formed by the reaction between solid C and excess Ti atoms. The main reactions in the Al-5Ti-0.62C system were that solid Al is transferred into liquid Al, and then liquid Al reacted with solid Ti to form the Al_3_Ti. At last, the release of a lot of heat promotes the formation of TiC which formed by the Ti atoms and solid C.

## 1. Introduction

Aluminum and its alloys have been widely used in military, aerospace, building, packaging, and electronics industries due to their excellent wear and corrosion resistance, low density, good high strength, high specific strength, and low efficiency of thermal expansion [1,2,3]. 

Nowadays, with the development of the industry, it is urgent to improve multi-properties for aluminum alloys [4,5]. Numerous studies have showed that the grain refining plays an important role in improving the properties of aluminum alloys [6,7,8]. Since Cibula [9] proposed the Carbide boride theory, tremendous efforts have been devoted to investigating the grain refining for aluminum alloy, including Al-Ti, Al-B, Al-C, Al-P, Al-Ti-B, Al-Ti-B-C, and Al-Ti-C master alloys [5,10,11,12]. Among which, Al-Ti-B and Al-Ti-C master alloys have been extensively studied at present. Furthermore, as an alternative grain refiner for Al-Ti-B master alloys, Al-Ti-C master alloys have the advantages of resistance to Zr poisoning and low aggregation tendency of TiC in aluminum melt [13,14,15,16]. Therefore, some works have been undertaken to investigate the influence of prepared processing parameters on phase synthesis and refinement performance of Al-Ti-C master alloys. Liu et al. [17,18] studied the effects of temperature and holding time on the microstructure of Al-Ti-C alloys. Yang et al. [19] and Gezer et al. [20] discussed the influence of the Ti/C stoichiometry on the grain refining performance of Al-Ti-C master alloys. Svynarenko et al. [21] investigated the importance of excess Ti for the refining efficiency of Al-Ti-C master alloy. Moreover, the in situ synthesis process of Al-Ti-C master alloys in aluminum melt by thermodynamic calculations and differential scanning calorimeter (DSC) experiment were analyzed [22,23,24,25]. It is considered that after the reaction of Ti with Al to form Al_3_Ti, the surplus Ti continues to react with Al_3_Ti to form the Ti-Al compound, and then reacts with C to form TiC at about 1273 K.

Obviously, the refining performance and phase distribution, morphologies, size of Al-T-C master alloys are seriously affected by the preparation process. Whereas, the mechanism of phases synthesis and refining are not uniform until now. Crossley and Mondolfo [26] proposed the peritectic theory according to the Al-Ti binary phase diagram. Wang et al [27] considered that the atom of Ti is provided by the dissolve of Al_3_Ti, and then moves to TiC, finally forming Al_3_Ti on the surface of TiC. Subsequently, peritectic reaction occurred on the surface of Al3Ti to form α-Al. Svynarenko et al [21] found that excess Ti can modify the growth and distribution of TiC.

This paper is supposed to investigate the process of the transformation of the phase and microstructure in the Al-5Ti-0.62C system by using the quenching experiment method. According to the thermodynamic calculation and DSC experiment, the synthesis mechanism of TiC in the Al-5Ti-0.62C system were researched. Particularly, this study provides a theoretical basis for the composition and process control of the Al-Ti-C master alloy. Besides, it also facilitates the preparation of high efficient and stable master alloy.

## 2. Experimental Materials and Methods

Al-5Ti-0.62C master alloy was prepared by the aluminum melt in-situ reaction, commercial pure aluminum (99.7 wt.%, A99.7), pure Al powder (99.0 wt.%, 80–100 μm), pure Ti powder (99.0 wt.%, 45–65 μm), and graphite powder (99.0 wt.%, 10–20 μm) were used as main raw materials (Tianjin Zhiyuan Chemical Reagent Co., Ltd, Tianjin, China). First, the powders were blended in the Pulaerisette-5 high-speed planetary ball mill (Nanjing Leibu Technology Industry Co., Ltd, Nanjing, China) at 350 r/min for 3 h, and the ball to material ratio is 3:1. Then, the mixed powders were pressed into a cylindrical preform in a stainless steel mold 25 mm in diameter and 50 mm in height on an AG-10TA universal test stretching machine (Shimadzu Corporation, Kyoto, Japan), and the uniaxial pressure was set to 50 Mpa. Next, the preform was preheated to 473 K in a drying oven. Simultaneously, a certain amount of A99.7 ingots were melted in alumina crucible by using a SG-7.5-10 type crucible furnace (Zhonghuan experimental furnace corporation, Tianjin, China), the temperature was heated to 1053 K. Subsequently, the preform was added into the A99.7 melt and held for 15 min to synthesize the Al-5Ti-0.62C master alloy, the flow chart is shown in Figure 1.

As shown in Figure 2, grain refining tests for the Al-5Ti-0.62C master alloy with different additions, i.e., 0%, 0.2%, and 0.3%, were carried out in order to evaluate its refining performance. After holding 5 min, the melts were stirred thoroughly by a graphite rod. Then, it was poured into a dye. After solidification, the specimens with the height of 1 cm were cut from the ingots and etched by cross-section by a reagent (45 mL HCl + 15 mL HNO_3_ + 15 mL HF + 25 mL H_2_O) within about 15–20 s. At last, pictures of the macro-structure were taken for each sample by a camera. 

Furthermore, the typical quenching samples at different stages of the reaction were prepared in order to investigate the phase and microstructure transition of the Al-5Ti-0.62C master alloy. At first, the preform immersed in the A99.7 melt was quickly taken out after reacting to different times (20, 45, 60, 90, 120 s) with control by paperless recorder, and then was quenched in a high pressure ice brine stream. After cooling, the quenching samples were cut along the axis and then coarsely ground, finely ground, mechanically polished, and finally, electrolytic polished with a reagent (10% HClO_4_ + 90% absolute ethanol, the composition of here is volume fraction, voltage is 20 V, at room temperature).

The phases in the quenched specimens and specimens for metallographic examinations were identified by D8 advance X-ray diffraction (XRD, Shimadzu Corporation, Kyoto, Japan, the tube has an accelerating voltage of 40 kV, an emission current of 40 mA, Cu Kα, λ = 1.54156 Å, scanning speed of 10°/min, step size of 0.02°, angle from 2 Theta 20° to 90°). The morphology and composition of the quenching samples and metallographic specimen were characterized by JSM-6700F scanning electron microscope (SEM, Shimadzu Corporation, Kyoto, Japan), an energy dispersive spectrometer (EDS, Shimadzu Corporation, Kyoto, Japan), and an electron probe microanalyzer (EPMA, Shimadzu Corporation, Kyoto, Japan). Further, the DSC curve of the Al-5Ti-0.62C system was carried out on NETZSCH STA 449F3 instrument (NETZSCH, Hanau, Germany). Air flow protection was used, and the heating rate was selected as 20 K/min, also, the weight of the sample was less than 10 mg. 

## 3. Results and Discussion

### 3.1. The Microstructure of Al-5Ti-0.62C Master Alloy

The XRD pattern and the SEM image of the Al-5Ti-0.62C master alloy are shown in Figure 3. It can be seen from Figure 3a that the main phases of the Al-5Ti-0.62C master alloy are α-Al, Al_3_Ti, and TiC. As shown in Figure 3b, the block-like particles with size of 5–8 μm are Al_3_Ti, and the little near spherical particles with size of 1–3 μm are TiC. Besides, not only the Al_3_Ti particles but also TiC particles are evenly distributed in the matrix, and TiC particles have not gathered together. 

### 3.2. Refining Effect Evaluation of Al-5Ti-0.62C Master Alloy

The macrostructures of A99.7 with different Al-5Ti-0.62C master alloy additions are shown in Figure 4. As shown in Figure 4a, A99.7 without refiner is mainly composed of surrounding coarse central equiaxed crystals and columnar crystals. When adding 0.2% Al-5Ti-0.62C into A99.7, the macrostructure of A99.7 is almost completely transformed into fine equiaxed crystals, except for fine columnar crystals located in the edge, as shown in Figure 4b. It is clearly observed from Figure 4c that A99.7 is entirely refined into the fine equiaxed structure when the amount of Al-5Ti-0.62C added increases to 0.3%.

The results above indicate that the Al-5Ti-0.62C master alloys exhibit excellent refining effect on A99.7. The main reason is that TiC can be used as a heterogeneous nuclear of α-Al [20,21,28]. According to the literature reports [29,30], α-Al and TiC all are face-centered cubic structure, lattice constant of TiC (a_TiC_) is 0.432 nm, and lattice constant of α-Al (a_Al_) is 0.404 nm, then the lattice mismatch (δ) between Al and TiC as Equation (1):(1)$$\delta \;=\;\frac{{\;(a}_{\mathrm{TiC}}{)\;-\;(a}_{\mathrm{Al}})}{a_{\mathrm{Al}}}\times \;100\%.$$

From the point of crystallization, TiC is beneficial to promote the nucleation of α-Al. Figure 5 shows the distribution of TiC in A99.7 after refining by the Al-5Ti-0.62C master alloy. The mapping analysis of the refining specimen with EPMA is shown in Figure 6. Figure 5a,b demonstrates that the TiC particles are mainly distributed in the core of the grain and aggregated on the grain boundary. From crystallographic considerations, in virtue of TiC and Al have a good orientation relationship, and the lattice mismatch between Al and TiC is 6.9%, which is less than 25% and larger than 5%. Namely, there is a semi-coherent interface between the TiC substrate and α-Al matrix, which can promote the heterogeneous nucleation ability of the α-Al matrix. Combined with the mapping analysis results, which are exhibited in Figure 6, the Ti elements mainly distribute around the TiC particles with a concentration gradient. Hence, Al_3_Ti will rapidly dissolve into Ti atoms and liquid Al when the Al-5Ti-0.62C master alloy was added into the Al melt, and then the dissolved Ti atoms will spread around the TiC particles to form the Ti concentration gradient, and finally form Al dendrite on the surface of the TiC particles by the Ti-rich zone, which demonstrate that TiC particles promote the α-Al nucleation. Furthermore, the amount of TiC particles is distributed at the grain boundary to limit the growth of the Al grains. 

The schematic diagram of the nucleation and growth process of α-Al after adding Al-5Ti-0.62C into the melt is shown in Figure 7. In the process of solidification, TiC particles are free in the melt at first, then the excess Ti atoms segregate around TiC to form a concentration gradient. Subsequently, some TiC particles are located in the grain, then Ti atoms, which generally form into a Ti-rich zone around TiC, promote the nucleation of dendrite α-Al. After solidification, Ti elements with the lace structure are wrapped inside the grain [27]. Some research explained that when the concentration of Ti reaches a certain level, a thin layer of Al_3_Ti will be formed on the TiC surface, and Al_3_Ti will react with the surrounding aluminum solution to form α-Al during the solidification process. In the subsequent cooling process, the first Ti-rich α-Al formed by the peritectic reaction was surrounded by the later Ti-poor α-Al formed. The central Ti was not easy to diffuse outward, so the composition gradient of Ti was formed in the α-Al, and the contrast was formed after erosion [31,32].

### 3.3. Phase Transformation and Microstructure Transformation of Al-Ti-C System

After adding the preform block into the A99.7 melt, the temperature of the preform block will increase rapidly, and liquid Al in the melt will infiltrate into the preform block. Then, the solid Ti will react with liquid Al to form Al_3_Ti. Also, solid Al particles melt into liquid Al [33]:3Al(l) + Ti(s) = Al_3_Ti(s) ΔG_1_,(2)
Al(s) = Al(l)           ΔG_2_.(3)

Subsequently, solid Ti particles are dissolved into Ti atoms and diffused into the Al melt. Afterwards, Ti atoms gather around solid C particles and react with C to form TiC. An amount of literature reports that TiC is mainly formed by the reaction between excess Ti or formed Al_3_Ti and solid C [20,34]:Ti(s) = [Ti]                  ΔG_3_,(4)
Al_3_Ti(s) + C(s) = TiC(s) +3 Al(l)   ΔG_4_,(5)
[Ti] + C(s) = TiC(s)          ΔG_5_.(6)

Furthermore, solid C in the Al melt also can be dissolved into C atoms and combined with Ti atoms to form TiC. Besides, solid C can react with Al to form Al_4_C_3_, and then Al_4_C_3_ will react with the Ti atom to form TiC [19,35], the reaction equations as follows: C(s) = [C]                   ΔG_6_,(7)
[Ti] + [C] = TiC(s)           ΔG_7_,(8)
4Al(l) + 3C(s) = Al_4_C_3_(s)        ΔG_8_,(9)
1/3 Al_4_C_3_(s) + [Ti] = TiC(s) +4/3 Al(l) ΔG_9_,(10)
Al_3_Ti(s) = 3Al(l) + [Ti](11)
where [Ti] and [C] present the dissolved Ti and C in the melt. The Gibbs free energy changes curves above formulas are shown in Figure 8. It can be seen that ΔG_1_ < 0, and ΔG_2_ < 0 when the temperature is larger than T1 (that is the melt point of Al, 933 K). In addition, ΔG_3_ to ΔG_5_ and ΔG_3_ to ΔG_9_ are also less than zero. In view of thermodynamics, Equations (2), (4)–(6) and (8)–(10) can be carried out spontaneously, and solid Al will spontaneously transfer into liquid Al when T ≥ 933 K. For Equation (7), ΔG_6_ > 0 until the temperature is equal to T4 (1550 K), that is, Equation (7) can be carried out spontaneously only when the temperature is greater than T4. In addition, Equation (2) will release a lot of heat, which will increase the local temperature rapidly. Y.F. Hou [36] researched that when the in-situ reaction of the aluminum melt occurs to prepare Al-Ti-C by quenching experiment, the local temperature will reach about 1600 K, which is consistent with the theoretical thermodynamic data in previous research. In other words, Equation (2) can promote the occurrence of Equations (7) and (8). However, the solubility of C in liquid Al is very low. The study of Dorward [37] shows that the dissolution of C in Al melt is about 10^−5^ wt.% at 973 K, and only about 3 × 10^−4^ wt.% at 1273 K. Therefore, from the thermodynamic point of view, the tendency of forming TiC in Equation (8) is lower than Equation (6). The possible compounds in the Al-Ti-C system are Al_3_Ti, Al_4_C_3_, and TiC, compared with ΔG_1_, ΔG_5_, ΔG_8_, and ΔG_9_, the value of the Gibbs free energy of Equation (10) is the most negative, so, TiC is the most stable phase, Al_3_Ti and Al_4_C_3_ will dissolve or react with other substances when T < T3 as in Equation (11). The stability of the phase is TiC > Al_3_Ti > Al_4_C_3_ in turn when T < T2, Banerji and Reif [38] believe that at 1273 K, C will not wet and react with Al, also, as shown in Figure 3, Al_4_C_3_ is not formed in Al-5Ti-0.62C master alloy, hence Equations (9) and (10) will not occur at the experiment temperature. As previously mentioned, TiC is likely formed by Ti atoms and solid C.

The reaction processes among Al, Ti, and C powders mixtures of the Al-5Ti-0.62C system identified by the DSC curve is shown in Figure 9. One endothermic peak and two exothermic peaks are shown in Figure 9. According to the above thermodynamic analysis and related literatures [39,40], and combined with the analysis of the Al-Ti binary phase diagram [33] and A-Ti-C ternary phase diagram [41,42], the first endothermic peak characterizes the melted of Al which corresponds to Equation (3) at 943.9 K. The first exothermic peak at 1193.9 K is presented by the formation of Al_3_Ti and is illustrated in Equation (2), the second exothermic peak at 1399.7 K is presented by the formation of TiC and is shown in Equation (6).

In order to study the phase transformation and microstructure transformation in the synthetic process of Al-5Ti-0.62C, the Al-5Ti-0.62C preform block was prepared and the microstructure under different reaction times in the melt was investigated. The XRD patterns of the quenched samples at different times are shown in Figure 10. It can be seen from Figure 10a,b that the main phases of the preform are α-Al, Ti, and C at the initial stage (20 s and 45 s). When the reaction time is 60 s, Al_3_Ti and TiC phases were detected by XRD as shown in Figure 10c. Furthermore, when the reaction time increases, the diffraction peaks intensity of Al_3_Ti and TiC phases gradually increase, as shown in Figure 10c–e. 

The SEM images and map scan patterns of the quenched sample at 120 s are shown in Figure 11. Table 1 shows EDS composition analysis of point A, point B, and point C in Figure 11. Figure 12 shows the line scan patterns of line 1 of Figure 11a. As can be seen from Figure 11, when the reaction time reaches 120 s, an amount of lace structure is presented in the melt as shown in Figure 11a. Combined with the results of the map scan patterns of Figure 11c–e, the center of the lace structure is the Ti particle, and Al, Ti elements are enriched at the edge of the lace structure. Further, Ti and C elements are surrounded by the lace structure. Moreover, the results of the EDS shows that Al:Ti of point A closely equals to 3:1, Ti:C of point B and C approximately equal to 1:1. A conclusion can be drawn that the block-like particles with the size of ~6 μm are Al_3_Ti at the edge of lace structure, and the particles with the size of ~1 μm are TiC around the lace structure. It can be seen from Figure 12 that the contents of the Ti elements from A to E show a decreasing tendency, and C elements mainly enrich in the AB stage and DE stage, the C element in the AB stage also has a tendency to spread outward, and in the DE stage, exiting in forms of TiC which is certified by EDS of point B and point C. 

The lace structure model of the Al-5Ti-0.62C system is shown in Figure 13. An amount of research reports that the wettability of Al and Ti is much higher than that of Al and C, therefore, liquid Al will rapidly spread on the surface of Ti particles after Al particles have melted, because the solubility of titanium in aluminum is very small, so it is rapidly saturated at the interface of the Al-Ti layer. Then, with the reaction of Al and Ti, the Al_3_Ti nucleus grows up to form Al_3_Ti particles. Moreover, because of the brittleness of Al_3_Ti, it is separated from the Al-Ti interface and entered into the Al solution under the continuous impact of liquid aluminum. Next, liquid aluminum flows into the gap of Al_3_Ti after separation, and continues to react with Ti to form Al_3_Ti, subsequently, Al_3_Ti breaks away from the Al-Ti interface and enters into the Al solution, and promotes the front Al_3_Ti particles to float away from the interface. In addition, the formation of Al_3_Ti will release a lot of heat, which will increase the local temperature to about 1600 K. Nevertheless, the melting point of Al_3_Ti is 1613 K, hence that it will cause the Al_3_Ti to fuse and break away from the Al-Ti layer. So, over and over again, until enough Al_3_Ti particles are distributed near the interface, the flow of liquid aluminum is blocked and the impact force is not enough to separate the Al_3_Ti particles from the titanium base wall. Subsequently, Al_3_Ti or Ti atoms provided by the Al_3_Ti diffusion reaction with the solid C form TiC particles.

The micro-kinetic model of TiC synthesis of the Al-5Ti-0.62C system is established by the analysis of the thermodynamic analysis of the Al-5Ti-0.62C system as shown in Figure 14. The TiC synthesis can be divided into three micro domains in the micro-region: the first micro-region is the generation of Al_3_Ti particles. When the preform block is put into the aluminum melt, the liquid aluminum in the aluminum melt penetrates into the preform block, so that the internal temperature of the prefabricated block is increased, afterwards, the aluminum powder is melted, and then the aluminum powder is wrapped on the surface of the Ti particles, hence, the Al-Ti layer is formed around the Ti particles by virtue of the solid–liquid diffusion, so the reaction I takes place. The second micro region is the dissolution of Al_3_Ti particles. As the reaction I is an exothermic reaction, the Al_3_Ti particles are separated from the Al-Ti layer as the reaction proceeds, and the Al_3_Ti particles undergo a dissolution reaction II under the action of high temperature, and the generated Ti is migrated to the third micro-region, and the generation of the TiC particles is provided [Ti]. The third micro-region is mainly the formation of TiC particles. The [Ti] of the second micro-region forms a titanium-rich layer around the C-particles, and the solid-state carbon reacts with Ti atoms provided the dissolved Al_3_Ti generates TiC particles. As the reaction proceeds, TiC is separated from the titanium-rich layer and free in the melt.

## 4. Conclusions

(1)Al-5Ti-0.62C master alloys have a good refining effect on A99.7. A99.7 is completely refined into the fine equiaxed structure by adding 0.3% Al-5Ti-0.62C master alloy.(2)When the Al-5Ti-0.62C master alloy is added into the melt, TiC particles mainly distribute in the grain interior and aggregate on the grain boundaries, excess Ti provided by the dissolved Al_3_Ti will spread around TiC and finally form a Ti-rich zone to promote the nucleation of α-Al.(3)From the view of thermodynamics and dynamics, TiC is formed by the reaction between solid C and Ti atom provided by the dissolved Al_3_Ti.(4)The main reactions in the Al-5Ti-0.62C system are confirmed by DSC, that is, solid Al is transferred into liquid Al, and then liquid Al reacts with solid Ti to form the Al_3_Ti, and release a lots of heat to promote the formation of TiC which is formed by the Ti atoms and solid C.

## Figures and Tables

**Figure 1 materials-13-00310-f001:**
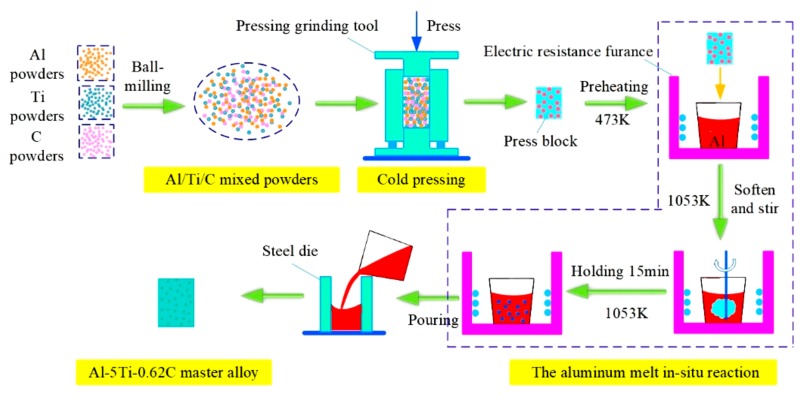
The experimental flow chart of the preparation of the Al-5Ti-0.62C master alloy.

**Figure 2 materials-13-00310-f002:**
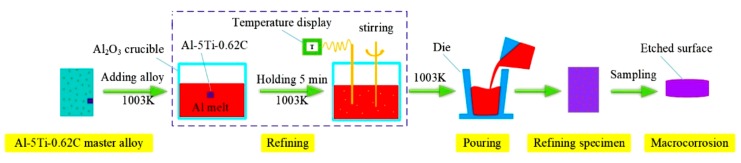
The experimental flow chart for refining commercial aluminum.

**Figure 3 materials-13-00310-f003:**
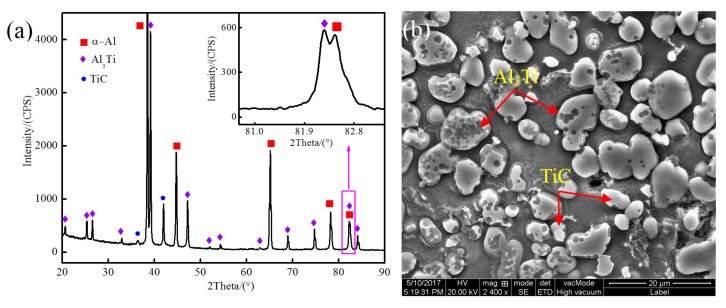
XRD pattern and SEM image of Al-5Ti-0.62C master alloy: (**a**) XRD pattern; (**b**) SEM image.

**Figure 4 materials-13-00310-f004:**
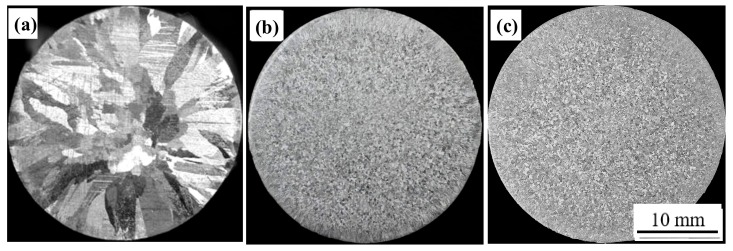
Macrostructures of A99.7 refined using the Al-5Ti-0.62C master alloy. (**a**) Without master alloy; (**b**) 0.2%; (**c**) 0.3%.

**Figure 5 materials-13-00310-f005:**
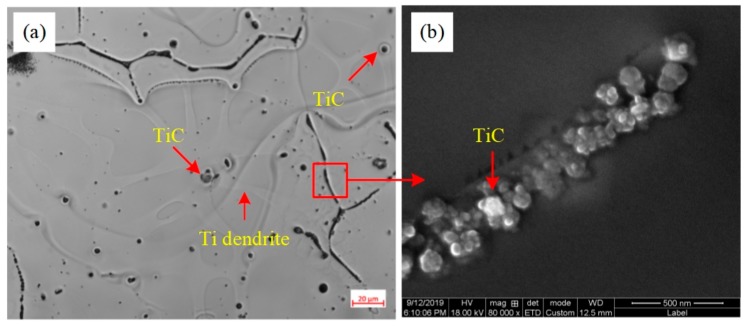
The grain morphologies of the distribution of TiC in A99.7 after refining: (**a**) OM image of A99.7; (**b**) SEM image on the grain boundary.

**Figure 6 materials-13-00310-f006:**
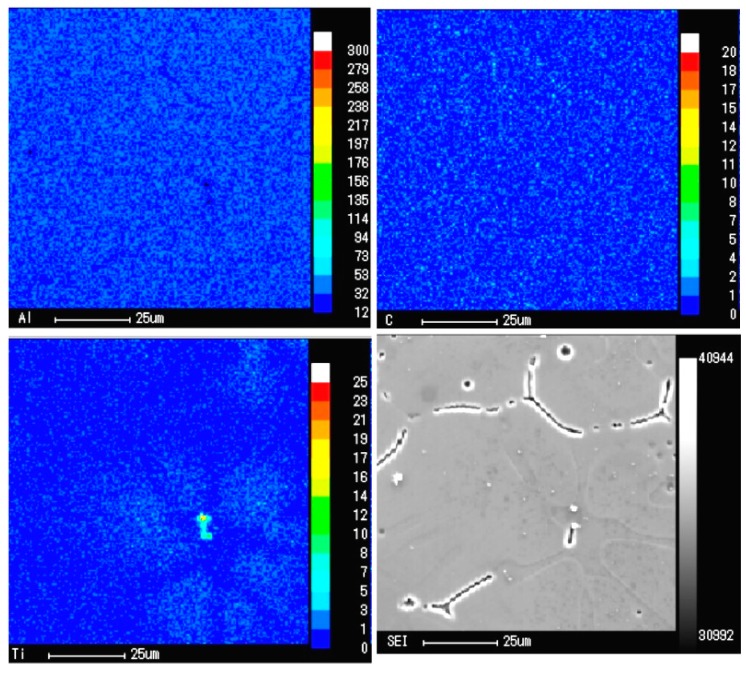
EPMA mapping analysis of A99.7 with the Al-5Ti-0.62C master alloy.

**Figure 7 materials-13-00310-f007:**
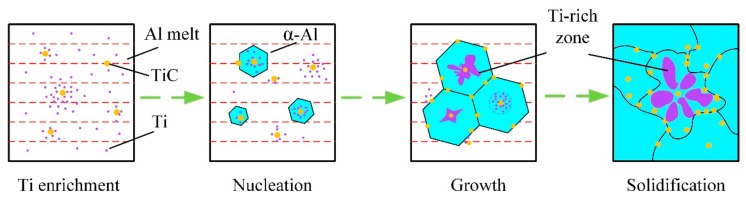
The schematic diagram of the nucleation and growth process of α-Al.

**Figure 8 materials-13-00310-f008:**
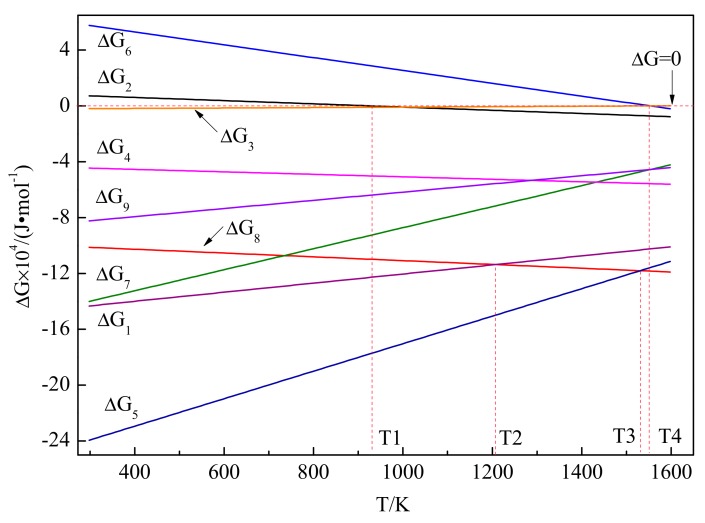
The Gibbs free energy changes curves of the reaction in the Al-5Ti-0.62C system.

**Figure 9 materials-13-00310-f009:**
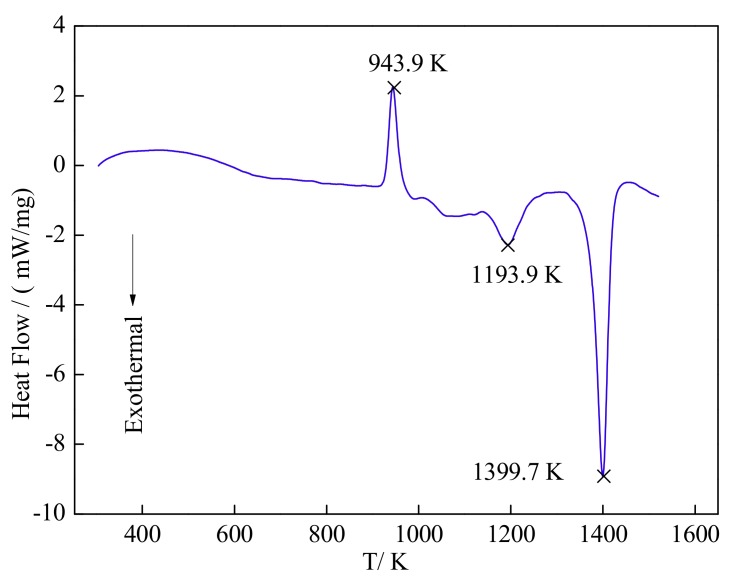
DSC curves of the Al-5Ti-0.62C system.

**Figure 10 materials-13-00310-f010:**
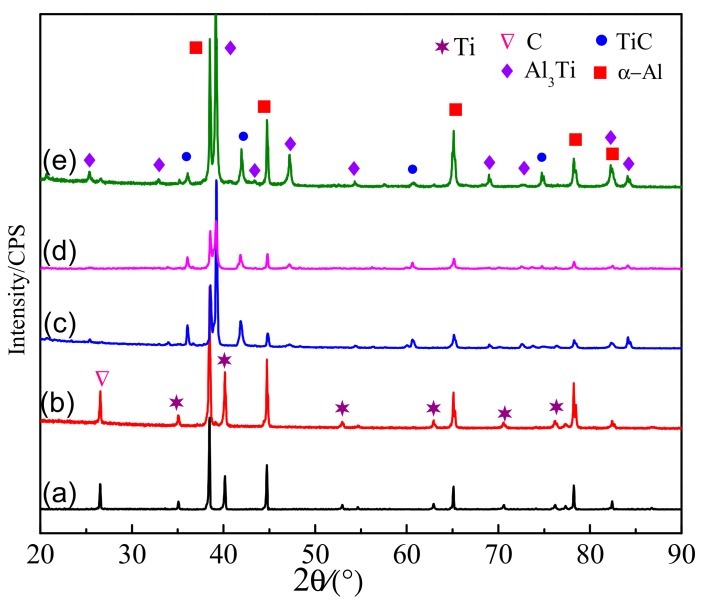
XRD patterns of the quenched samples with different reaction times: (**a**) 20 s; (**b**) 45 s; (**c**) 60 s; (**d**) 90 s; (**e**) 120 s.

**Figure 11 materials-13-00310-f011:**
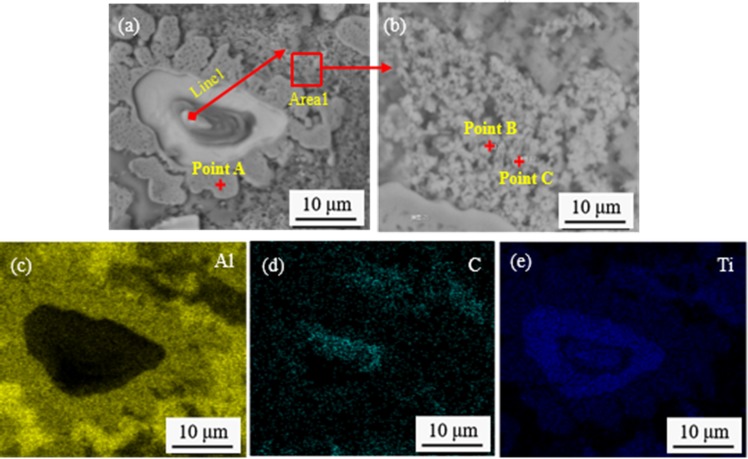
SEM images and map scan patterns of the quenched sample at 120 s: (**a**) SEM image; (**b**) SEM image of Area 1; (**c**–**e**) map scan patterns of Al, C, and Ti, respectively.

**Figure 12 materials-13-00310-f012:**
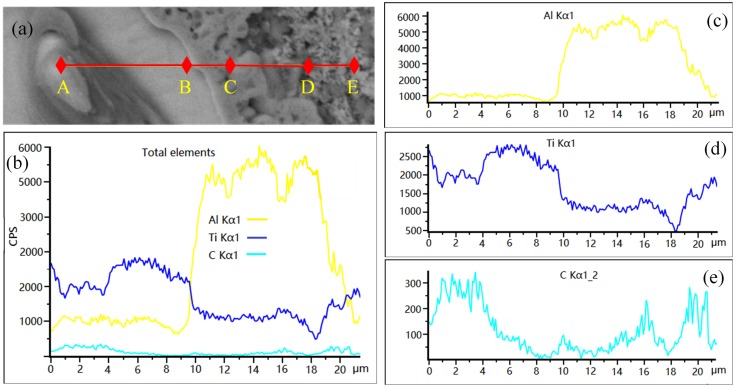
SEM image and line scan patterns of Line 1 of Figure 11a: (**a**) SEM image; (**b**) line scan patterns of all elements; (**c**–**e**) line scan patterns of Al, Ti, and C, respectively.

**Figure 13 materials-13-00310-f013:**
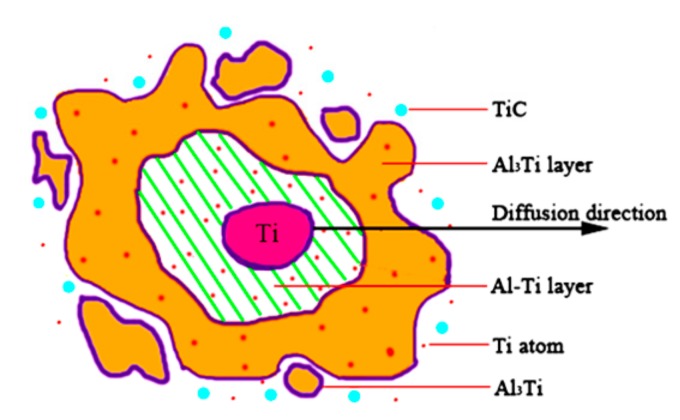
The lace structure model of the Al-5Ti-0.62C system.

**Figure 14 materials-13-00310-f014:**
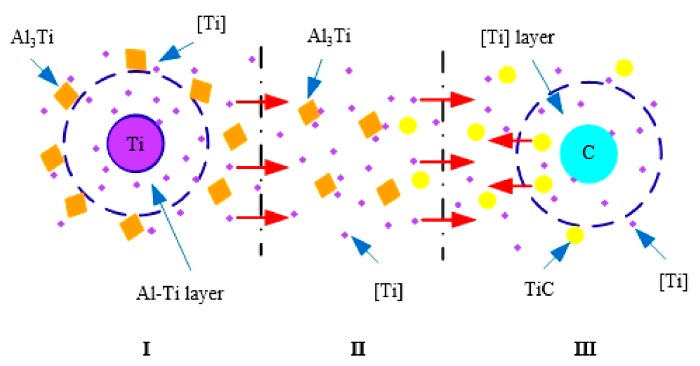
Microscopic kinetic model of TiC synthesis in the Al-5Ti-0.62C system. (**I**) 3Al(l) + Ti(s) → Al_3_Ti(s), (**II**) Al_3_Ti(s) → [Ti] + 3Al(l), (**III**) [Ti] + C(s) → TiC(s).

**Table 1 materials-13-00310-t001:** The energy dispersive spectrometer (EDS) composition analysis of point A, point B, and point C in Figure 11b.

Point No.	Atomic (Al)/%	Atomic (Ti)/%	Atomic (C)/%
A	70.6	25.7	3.6
B	13.2	41.7	45.2
C	10.4	40.5	49.1

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
