# Peer review of "Investigation of Microstructure of Al-5Ti-0.62C System and Synthesis Mechanism of TiC"

_materials, 2020, doi:10.3390/ma13020310_

Round 1
Reviewer 1 Report
Introduction - see line 36 capital letter after comma. Introduction - provides an overview of the existing viewpoints on the grain refinement of Al by using Al-Ti-C master alloys. Together with consideration of TiC effectiveness as nucleation sites for alfa-Al grains, it will be better to add certain remarks on the stability of TiC particles in Al melt, as an example - see Rapp R A, Zheng X. Thermodynamic consideration of grain refinement of aluminium alloys by titanium and carbon. Metall Trans A, 1991, A22: 3071-3075; Page 3, line 91 - "microstructure translation" - microstructure transition; Page 3, line 92 CP Al - it would be better to specify A99.7 or other; Page 3, line 97 - would better to specify the temperature of electrolyte (room temperature or lower temperature, normally such electrolytes used at lower than the room temperature); Page 3, line 98 - "The phase of quenching samples and metallographic specimen...." - The phases in quenched specimens and specimens for metallographic examinations....this would be better Page 6, line 157 - "gradient. Subsequently, some TiC particles are located in the grain, Then enrich Ti atoms 157 which around TiC generally formed into a Ti dendrite to promote the nucleation of α-Al" - capital letter after comma, and from my point of view Ti dendrites do not form and authors mentioned that the exist certain enrichment of Ti in the central part of grain, what is obvious because of the strong ability of Ti to segregate during solidification of Al. Page 7, line 174 - "Where [Ti] and [C] present dissolved the Ti and C in the melt. - what is the sense of this sentence? Page 7, line 187 - "Therefore, from a dynamic point of view, the..." - probably "from the thermodynamic point of view..."; Page 7, line 193 - "will not witting and react with Al.." - witting is not the appropriate word - wetting would give the sense to the sentence; Conclusive remark - to improve the quality of the article it would be better to add microstructure of separate a-Al grain where TiC particles are located exactly at the centre of the grain. It is clear if the particle (particles) nucleate a-Al it will be positioned in the grain centre. From the images of authors, it is not clear whether TiC particles act as nucleation sites for a-Al grain.Author Response
Dear expert:
Thank you very much! I feel very honored for that you have revised my manuscript and given good comments again. I agree with your recommendations and have carefully revised it according to your suggestion. I provide a cover letter to explain “point by point” the details of the revisions in the manuscript and this is my reply as follows:
Comments and Suggestions for Authors
Point 1: Introduction - see line 36 capital letter after comma.
Response 1: Thank you very much for your suggestion. I agree with your recommendation and have revised the letter. Please see the red label on line 36 in revised manuscript.
Point 2: Introduction - provides an overview of the existing viewpoints on the grain refinement of Al by using Al-Ti-C master alloys. Together with consideration of TiC effectiveness as nucleation sites for alfa-Al grains, it will be better to add certain remarks on the stability of TiC particles in Al melt, as an example - see Rapp R A, Zheng X. Thermodynamic consideration of grain refinement of aluminium alloys by titanium and carbon. Metall Trans A, 1991, A22: 3071-3075;
Response 2: Thank you very much for your valuable comment and suggestion for my paper. I understand your idea very well. But this paper is main to study the phase transition and microstructure transition during the preparation of Al-Ti-C master alloy. The study on the refining mechanism of industrial pure aluminum by Al-Ti-C master alloy is simply described. Many studies have been reported about the the grain refinement of Al by using Al-Ti-C master alloys, and consideration of TiC effectiveness as nucleation sites for α-Al grains. Our later research will around this direction to further in-depth discussed.
Point 3: Page 3, line 91 - "microstructure translation" - microstructure transition;
Response 3: Thank you very much for your suggestion. I agree with your recommendation and have modified "microstructure translation" to "microstructure transition" in the manuscript. Please see the red label on line 91 in the revised manuscript.
Point 4: Page 3, line 92 CP Al - it would be better to specify A99.7 or other;
Response 4: Thank you very much for your suggestion. I agree with your recommendation and have specified CP Al as A99.7 on page 2, line 68, and then i have modified CP Al to A99.7. Please see the red label in the revised manuscript.
Point 5: Page 3, line 97 - would better to specify the temperature of electrolyte (room temperature or lower temperature, normally such electrolytes used at lower than the room temperature);
Response 5: Thank you very much for your suggestion. I understand your idea very well. In this paper, eletrolytes used at room temperature, and i have specified the temperature in the manuscript. Please see the red label in the revised manuscript.
Point 6: Page 3, line 98 - "The phase of quenching samples and metallographic specimen...." - The phases in quenched specimens and specimens for metallographic examinations....this would be better
Response 6: Thank you very much for your valuable comment and suggestion for my paper. I understand your idea very well. I have complemented the description of the sentence in the manuscript. Please see the red label in the revised manuscript.
Point 7: Page 6, line 157 - "gradient. Subsequently, some TiC particles are located in the grain, Then enrich Ti atoms 157 which around TiC generally formed into a Ti dendrite to promote the nucleation of α-Al" - capital letter after comma, and from my point of view Ti dendrites do not form and authors mentioned that the exist certain enrichment of Ti in the central part of grain, what is obvious because of the strong ability of Ti to segregate during solidification of Al.
Response 7: Thank you very much for your valuable comment and suggestion for my paper. I understand your idea very well. I have revised the word after comma into ''then''. And then i looked up some literature, like ref. 31 and ref. 32, found that Ti-rich zone formed around TiC, it can promote the form of dendrite α-Al, which was surrounded by the later Ti-poor α-Al formed, but the specific formation mechanism and the role of Ti-rich zone during refinment mechanism need to be further studied. About some explanations can be seen in the revised manuscript, i hope you will be satisfied.
Point 8: Page 7, line 174 - "Where [Ti] and [C] present dissolved the Ti and C in the melt. - what is the sense of this sentence?
Response 8: Thank you very much for your suggestion. This sentence is main to explain the existed state Ti and C, [Ti] and [C] is mean to the atomic status, which different from the solid.
Point 9: Page 7, line 187 - "Therefore, from a dynamic point of view, the..." - probably "from the thermodynamic point of view...";
Response 9: Thank you very much for your valuable comment and suggestion for my paper. I agree with your recommendation and have modified the sentence in the manuscript. Please see the red label on page 7, line 192 in the revised manuscript.
Point 10: Page 7, line 193 - "will not witting and react with Al." - witting is not the appropriate word - wetting would give the sense to the sentence;
Response 10: Thank you very much for your valuable comment and suggestion for my paper. I understand your idea very well. I have revised the sentence to express the right meaning, that is, C will not wet and react with Al. Please see the red label on page 7, line 197 in the revised manuscript.
Point 11: Conclusive remark - to improve the quality of the article it would be better to add microstructure of separate a-Al grain where TiC particles are located exactly at the centre of the grain. It is clear if the particle (particles) nucleate a-Al it will be positioned in the grain centre. From the images of authors, it is not clear whether TiC particles act as nucleation sites for a-Al grain.
Response 11: Thank you very much for your suggestion. But i think it can be obviously seen from Fig. 5 and Fig. 6 that TiC particles existed in the center of α-Al, that is, TiC is the heterogenous core of the grain, in particular, which can be clearly seen in figure 6. In addition, we will carry out a more in-depth and detailed study on the interface between TiC particles as nucleation core and Al matrix in the subsequent research part about refinement mechanism. I hope you will be satisfied.

Reviewer 2 Report
In my opinion, it is a really nice and high-quality paper suitable for its publication in Materials. A lot of interesting experimental results blend here with its deep enough interpretations. The most interesting part to my mind is 3.3 Phase transformation and microstructure transformation of Al-Ti-C system. My main concern here is the number of typos throughout the text. Among them: Abstract, line 16: "the main phase.. are" Introduction, line 36: "...,Tremendous" line 159: "...are for in the grain interior [27]." line 176: apparently, must be G2 here instead of G1 line 177: the constructions like "dG3 to dG5" is not clear for me Figure 12: all labels and titles are quite small and one of them in Chinese lines 202,203: it should be useful to put here more detailed explanations why these temperatures correspond to certain reactions dG with their equations. I strongly recommend the authors to read the manuscript carefully again and correct all typos in the text simultaneously improving less clear sentences in the discussion.Author Response
Dear expert:
Thank you very much! I feel very honored for that you have revised my manuscript and given good comments. I provide a cover letter to explain “point by point” the details of the revisions in the manuscript and this is my reply as follows:
Comments and Suggestions for Authors
In my opinion, it is a really nice and high-quality paper suitable for its publication in Materials. A lot of interesting experimental results blend here with its deep enough interpretations. The most interesting part to my mind is 3.3 Phase transformation and microstructure transformation of Al-Ti-C system. My main concern here is the number of typos throughout the text.
Thank you very much for your valuable comments and suggestions for my paper. I agree with your recommendations. I have carefully revised it according to your suggestion. Please see revised article and the reply below.
Point 1: Abstract, line 16: "the main phase.. are"
Response 1: Thank you very much for your suggestion. I agree with your recommendation and have modified the sentence in the manuscript. Please see the red label on page1, line 16 in the revised manuscript.
Point 2: Introduction, line 36: "...,Tremendous"
Response 2: Thank you very much for your suggestion. I agree with your recommendation and have revised the letter. Please see the red label on line 36 in revised manuscript.
Point 3: line 159: "...are for in the grain interior [27]."
Response 3: Thank you very much for your suggestion. I agree with your recommendation and have modified the sentence in the manuscript. Please see the red label on page6, line 163 in the revised manuscript.
Point 4: line 176: apparently, must be G2 here instead of G1
Response 4: Thank you very much for your suggestion. I agree with your recommendation and have modified G1 into G2. Please see the red label on page7, line 187 in the revised manuscript.
Point 5: line 177: the constructions like "dG3 to dG5" is not clear for me
Response 5: Thank you very much for your suggestion. dG3 to dG5 is mean to the the Gibbs free energy changes of Eq.(4), Eq.(5)and Eq.(6).
Point 6: Figure 12: all labels and titles are quite small and one of them in Chinese
Response 6: Thank you very much for your suggestion. I agree with your recommendation and have modified all labels and titles, and changed from Chinese to English in Figure 12. Please see the red label on page11, line 252 in the revised manuscript.
Point 7: lines 202,203: it should be useful to put here more detailed explanations why these temperatures correspond to certain reactions dG with their equations.
Response 7: Thank you very much for your suggestion. I agree with your recommendation and have added some references to certify the temperatures correspond to certain reactions dG. In addition, combined with the pahse digram to prove the reaction.
Point 8: I strongly recommend the authors to read the manuscript carefully again and correct all typos in the text simultaneously improving less clear sentences in the discussion.
Response 8: Thank you very much for your suggestion. I agree with your recommendations and have carefully correct typos in the text, and simultaneously improved less clear sentences. Please see the red label in the revised manuscript, i hope you will be satisfied.
